# Generative Retrieval with Large Language Models: Fine-Grained Self-Recall

## Abstract

Knowledge-sensitive NLP tasks require access to a large volume of world or domain knowledge. Previous methods all require an extra retrieval model to obtain a related reference passage for answering. However, this paper finds that a large language model itself can generate an existing passage solely based on the question through constrained decoding, thereby achieving a retrieval effect and enhancing prediction. We propose a two-stage method, **LLM2GR**. Specifically, we first prompt the large language model to generate relevant document title identifiers in a constrained manner. We then prompt it to generate a passage within the document set selected in the first stage and choose the final reference passage through scoring weighting of the two stages. To speed up the generation retrieval, we only generate a shorter prefix rather than a complete passage, then locate it in the document to extract a longer, complete reference passage. This method requires no additional retrieval models, no extra training, and no advance text chunking, and can be applied to documents of any length. Experiments on 6 KILT benchmark knowledge-sensitive tasks have verified the effectiveness of our method.

## 1 Introduction

Knowledge-intensive tasks, including open-domain question answering, dialogues, and fact checking, require access to considerable world or domain-specific knowledge (Petroni et al., 2021). Common approaches involve utilizing external knowledge sources such as Wikipedia and using additional sparse or dense retrieval models to initially retrieve a few relevant context passages from Wikipedia, and then predict the answer under the condition of the question (Karpukhin et al., 2020; Lewis et al., 2020; Izacard & Grave, 2021). However, these traditional retrieval methods exhibit several drawbacks. First, the candidate documents used for retrieval are divided into chunks (e.g., 100 words), and the segmented part is prone to some information loss. Second, in modern dual-tower dense retrieval models, the representations of questions and documents are usually obtained independently (Karpukhin et al., 2020), leading them to only capture shallow interactions (Khattab et al., 2021). And the additional models can't take advantage of the world knowledge or reasoning ability of large language models (Levine et al., 2022).

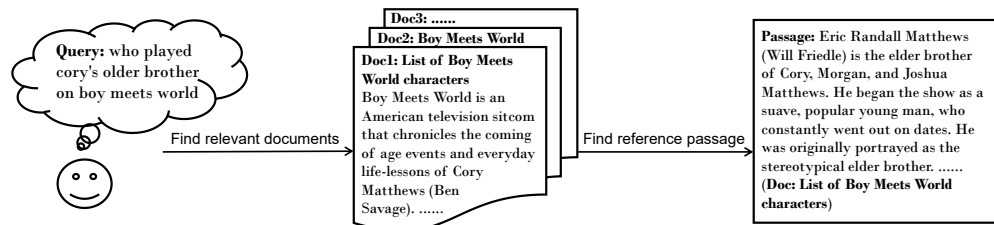

Figure 1: A common process for humans to search for information often involves an intermediary step of locating document pages, instead of directly finding the reference passage.

Compared with traditional sparse and dense retrieval, generative retrieval methods (Cao et al., 2021; Tay et al., 2022; Bevilacqua et al., 2022; Wang et al., 2022; Lee et al., 2022) have gained increasing attention. They generate document identifiers or documents themselves through auto-regressive

generative models that execute deep token-level cross-attention, interacting with the entire parameter space of models that are trained on the target corpus to overcome bottlenecks. But previous methods usually required a lot of training so they are only applied to smaller language models. Recently, Ziems et al. (2023) have proposed using the large language model GPT-3 to retrieve documents by generating document URLs. However, this method can only be applied to documents retrievable by URLs and can locate only at the page-level, not at a finer passage-level.

In this paper, we find that large language models can mimic the process humans use to search for information: first finding relevant documents or pages, then locating the specific reference passage within them, as shown in Figure 1. We propose a two-stage generative retrieval method named **LLM2GR**. Specifically, in the first stage, we prompt the large language model to generate short title identifiers and use prefix-tree (Trie) (Cormen et al., 2022) constrained decoding to ensure that the generated titles are all within the knowledge source. In the second stage, using the top-ranking documents obtained from the first stage, we construct a new FM-index (Ferragina & Manzini, 2000) that can effectively recognize any substring within the set. Subsequently, we prompt the large language model to generate the relevant passages needed for the problem and ensure that the generated passages are within the document set through FM-index constrained decoding. To leverage the information generated in both stages, we use a weighted sum of the scores from the two-stage generation to select the final retrieved passage.

Although large language models boast impressive capabilities, the process of generating complete passages can be overly time-consuming, posing a significant drawback to their practical implementation. To address this issue, we propose a novel approach termed Short Prefix Generation and Location (SPGL). This method commences with the generation of a shorter prefix, followed by locating the document that includes this prefix in the document set obtained from the first stage. Subsequently, we use the Knuth-Morris-Pratt (KMP) algorithm to identify the start position of the prefix in the document, and from this position, we determine a longer passage as the final retrieval result. This approach substantially accelerates the retrieval speed of large language models. Our method offers novel insights into further harnessing the world knowledge stored in large language models, making them adaptable to a variety of knowledge-intensive tasks.

We conduct extensive experiments on 6 KILT benchmark (Petroni et al., 2021) knowledge-sensitive tasks. In a zero-shot setting, solely through the open-source large language models Llama (Touvron et al., 2023a) or Llama2 (Touvron et al., 2023b), without the need for any additional retrieval models and without pre-chunking the documents, our method can achieve better page and passage-level retrieval results compared to traditional methods with additional models and enhance the performance of downstream tasks.[1]

In summary, our main contributions are as follows:

1. We utilize large language models for the first time to generatively retrieve fine-grained passages in a zero-shot setting, without extra training, retrieval models, or text chunking.

2. We introduce a two-stage method, generating retrieval title identifiers first and then retrieval passages, with the final reference passage determined by a weighted score from both stages.

3. By generating shorter prefixes and using the KMP algorithm for locating passage positions, we significantly speed up the generative retrieval in large language models.

4. Across 6 knowledge-sensitive NLP tasks, our method excels in page and passage-level retrieval and significantly enhances downstream task performance with the retrieved passages.

## 2 RELATED WORK

### 2.1 RETRIEVAL-THEN-READING APPROACH

Knowledge-intensive NLP tasks usually implement a retrieval-then-read model pipeline. Firstly, a retrieval model filters potential relevant passages from a vast database (e.g., Wikipedia) in response to a question. A reader then scans these passages for the answer. Current research tends to enhance either the retrieval (Karpukhin et al., 2020; Khattab & Zaharia, 2020; Qu et al., 2021; Izacard et al., 2022) or reader components (Izacard & Grave, 2021; Cheng et al., 2021; Yu et al., 2022), or develops end-to-end systems (Lewis et al., 2020; Singh et al., 2021). Traditional techniques like TF-IDF and

---

[1]Code is available at https://anonymous.4open.science/r/LLM2GR-84E6/.

BM25 utilize sparse retrieval, matching question and text chunks (Robertson et al., 2009; Chen et al., 2017; Yang et al., 2019). Recent methods e.g., ORQA (Lee et al., 2019) and DPR (Karpukhin et al., 2020), use dense context vectors for indexing documents, enhancing performance. However, in dual-encoder dense retrieval models, question and document representations are obtained independently, creating performance limitations due to the shallow vector interaction (Khattab & Zaharia, 2020). We propose a new strategy, leveraging the internal parameter interactions of large language models for both retrieval and reading, without the need for additional models and training steps.

## 2.2 GENERATIVE RETRIEVAL

Interest has surged in employing autoregressive language models to generate identifier strings, simplifying the retrieval process and addressing the limited interaction bottleneck in dual-encoder models. For example, Cao et al. (2021) used Wikipedia page titles for retrieval, Tay et al. (2022) targeted generation as root-to-leaf paths in a hierarchical clustering tree and Bevilacqua et al. (2022) mapped to distinctive n-grams. Recently, Lee et al. (2022) proposed a generative multi-hop retrieval method, Li et al. (2023b) employed multiple identifiers collaboratively to determine retrieval passages, and Ren et al. (2023) introduced a two-stage method that first generates passages, followed by generating URL identifiers. Despite their achievements, their intensive training makes it hard to apply in large language models. To leverage these models' generative capacity, some methods use generation for query expansion (Mao et al., 2021; Gao et al., 2022), while others generate document URLs, followed by traditional retrieval techniques (Ziems et al., 2023). Yet, these still require additional retrieval models and utilize proprietary large language models. Our proposed method uniquely leverages the capacity of large language models for zero-shot, page-level, and fine-grained passage-level retrieval, achievable with only a 13B or even 7B open-sourced large language model.

## 2.3 LARGE LANGUAGE MODEL OUTPUT ENHANCING NLP MODELS

Recent studies have discovered that relevant knowledge can be extracted from large language models through prompting, especially in areas where knowledge bases are inadequately covered (Liu et al., 2022; Fang et al., 2022). Enhancing model performance by generating outputs with large language models has also garnered attention. For instance, chain-of-thought learning has been introduced, focusing on prompting large language models to generate sequences of intermediary reasoning steps (Wei et al., 2022; Kojima et al., 2022; Li et al., 2022). Trivedi et al. (2023) utilized this generated chain of thought to guide the external retrieval process. On the other hand, Liu et al. (2022); Sun et al. (2023); Yu et al. (2023) proposed using GPT-3 to generate related contexts for "retrieval", incorporating these contexts as additional input when answering questions. However, the full generation of contexts through large language models inherently still suffers from the hallucination phenomenon (Li et al., 2023a). Moreover, generating complete contexts is both time-consuming and expensive. Our method, by generating existing document passages, ensures the absence of hallucinations and, by only producing shorter prefixes, enhances the cost efficiency and speed of large language models in generating retrieval-relevant contexts.

## 3 PROPOSED METHOD

In this section, we detail our two-stage method **LLM2GR**. In the first stage, we prompt a large language model to generate retrieval title identifiers, used as candidate documents for the next stage. The second stage involves prompting the model to retrieve passages from the documents obtained in the first stage. To enhance retrieval speed, we generate only shorter prefixes and extract the retrieved passages from the located positions in the documents. The structure of our method is shown in the Figure 2.

### 3.1 FIRST STAGE: ZERO-SHOT AUTOREGRESSIVE TITLE IDENTIFIER RETRIEVAL

When faced with knowledge-sensitive tasks, similar to the human thinking process, the model initially needs to accurately consider corresponding documents, such as Wikipedia pages, which contain ample information that can be used to answer questions. LLM-URL (Ziems et al., 2023) utilized large language models to directly generate page URLs for location. Since smaller large language models have relatively inferior understanding of prompts, directly generating URLs may result in

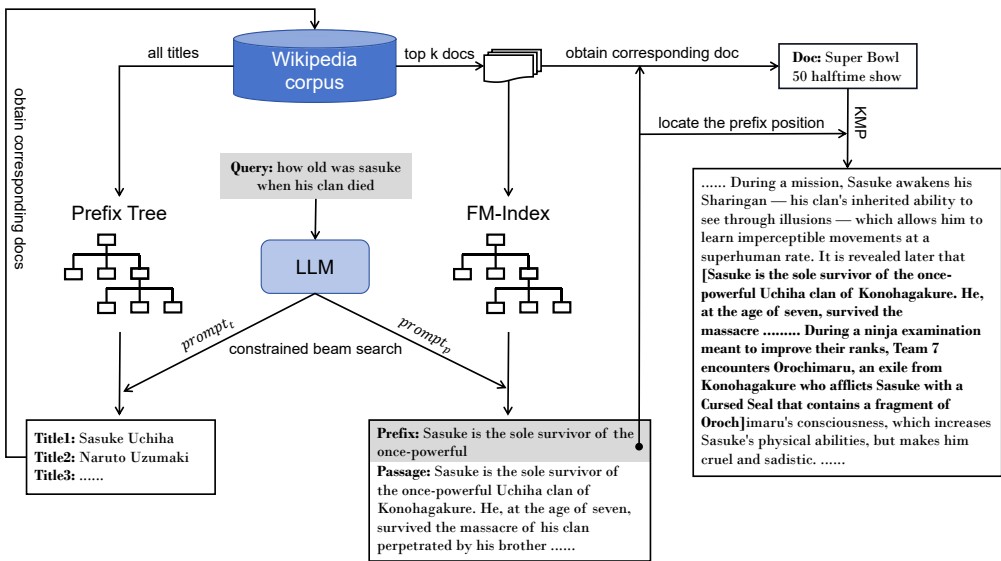

Figure 2: The **LLM2GR** method's architecture is depicted. Initially, all Wikipedia titles are stored in a prefix tree, and the Large Language Model (LLM) is prompted to generate title identifiers, retrieving corresponding documents. Subsequently, an FM-index is constructed from the top $k$ documents, and the LLM generates passages under this constraint. The gray section represents the generated prefix, used to locate the corresponding document and extract the full passage with the KMP algorithm.

numerous non-existent URLs, thereby affecting the outcome of retrieval. Inspired by their research, for Wikipedia page documents with titles, we can directly prompt the large language model to generate existing title identifiers and uniquely determine the title through constrained search. This method utilizes the knowledge stored within the large language model and can also ensure that the generated title is uniquely existing.

For the input query $x$, in the first stage of our proposed method, to prompt the model to generate corresponding Wikipedia titles, we utilize a prompt $prompt_t(x)$ to stimulate the language model to generate potential Wikipedia titles that could cover the query's content. For instance, for open-domain question-answering tasks, we utilize the prompt: "Question: {}\n\nThe Wikipedia title corresponding to the above question is:\n\nTitle:".

We define the set of all Wikipedia titles as $T$, and the set of all documents as $D$, such that every title and document uniquely correspond to each other. First, we store all Wikipedia titles $T$ in a prefix tree (Trie) (Cormen et al., 2022). At each step in the beam search of the large language model, we ascertain the set of tokens to be generated next by making use of both the prefix tree and the previously generated tokens, and mask the logits of tokens not belonging to this set as $-\infty$. In other words, the prefix tree acts as a navigation structure, guiding the model towards generating tokens following a path corresponding to a known title $t$ in the set $T$. A detailed introduction to Trie can be found in Appendix A.1.1. We compute the autoregressive generation's score through the default implementation in the (Wolf et al., 2020) library, with the score of title $t$ given $prompt_t(x)$:

$$score_1(t|prompt_t(x)) = \frac{logp_\theta(y_t|prompt_t(x))}{|y_t|} = \frac{\sum_{i=1}^{l_t} logp_\theta(y_i|y_{<i}, prompt_t(x))}{l_t}, \quad (1)$$

where $y_t$ represents the set of tokens in the title $t$, $l_t$ and $|y_t|$ represent the number of tokens generating title identifiers, $\theta$ is the model's parameters.

### 3.2 SECOND STAGE: ZERO-SHOT AUTOREGRESSIVE PASSAGE RETRIEVAL

Given the substantial length of text in most pages, retrieved documents cannot be directly used as context input for large language models. Therefore, we need a method to extract specific segments

relevant to a given query. Traditional retrieval techniques often involve text chunking for lengthy documents, but this approach can lead to information loss and may not always perform satisfactorily with extensive texts. To overcome this challenge, we adopted a more direct method for extracting passages, utilizing large language models to generate passages directly from the pages. We activate the language model to generate a passage $p$ containing pertinent information using the prompt $prompt_p$, for example: "Question: \n\nThe answer to the above question can be found in the following Wikipedia paragraph:\n\nAnswer:".

However, allowing large language models to generate content directly often results in the production of passages not originally present in the documents, with no guaranteed format, and the emergence of hallucination phenomena (Li et al., 2023a), thereby affecting downstream performance. To ensure that the generated passages are part of the Wikipedia pages retrieved in the first stage and to guarantee the accuracy of the generated passages, we employ constrained beam search for generation. Firstly, we construct a new FM-index (Ferragina & Manzini, 2000) for the document set $D_k$ corresponding to the top $k$ title identifiers obtained in the first stage. The FM-index can be considered a specialized prefix tree that supports searches starting at any position. Given a starting token or string, the FM-index can provide a list of possible token successors in $O(V log(V))$ time, where $V$ is the size of the vocabulary. A detailed introduction to FM-index can be found in Appendix A.1.2. We determine the set of allowable tokens for the subsequent generation based on the part generated earlier, enabling us to generate complete text passages $p$ from any section of the document set $D_k$. We measure the likelihood of the generated text passages by calculating scores for them using an autoregressive formula:

$$score_2(p|prompt_p(x)) = \frac{logp_\theta(y_p|prompt_p(x))}{|y_p|} = \frac{\sum_{i=1}^{l_p} logp_\theta(y_i|y_{<i}, prompt_p(x))}{l_p}, \quad (2)$$

where $y_p$ represents the set of tokens in passage $p$, $\theta$ are the model parameters, and $|y_p|$ and $l_p$ denote the number of tokens generated in the passage, which is typically set between 150 and 200. To integrate the information generated in both stages, we calculate the weighted sum of the scores from the first and second stages to obtain the final passage score under the input query $x$:

$$score(p|x) = \alpha * score_1(t|prompt_t(x)) + (1 - \alpha) * score_2(p|prompt_p(x)), \quad (3)$$

where $\alpha$ is a hyperparameter, and $score_1(t|prompt_t(x))$ is the score corresponding to the generated passage $p$'s Wikipedia title $t$. Consequently, the passage with the highest score is chosen as the reference passage.

### 3.3 SHORT PREFIX GENERATION AND LOCATION

Despite having a powerful model capable of generating long passages, its expensive inference speed to an extent undermines its practicality in generative retrieval. In fact, to generate a passage with a length of around 150 to 200 tokens, a considerable amount of computational resources and time are invested, which is intolerable in many real-world scenarios. Note that aside from the size of beam search, the length of generation is a crucial factor impeding the generation speed of large language models. To tackle this challenge, we propose a novel method — Short Prefix Generation and Location (SPGL). The fundamental idea of SPGL is to firstly generate a relatively short text prefix, and following that, locate and extract a complete passage that contains this prefix from the source document set. In specific, we split the whole process into two major steps as follows.

Initially, given an input question $q$, we prompt a large language model to generate a short text prefix $p_s$ of $l_{p_s}$ tokens guided by a prompt $prompt_p$, which's identical to the one used in the second stage. In this step, we set $l_{p_s}$ to be significantly less than the full length of the long text, thus remarkably saving generation time and computational resource.

Subsequently, we find the document $d$ in the first stage obtained document set $D_k$ that corresponds to $p_s$. Since we have controlled the number of documents in the document set $D_k$ obtained in the first stage, in the vast majority of cases, we can obtain a unique document $d$ containing $p_s$. When we are unable to obtain a unique document, we default to selecting the first document. In the next step, we use the Knuth-Morris-Pratt (KMP) string matching algorithm to quickly determine the start position $st$ of $p_s$ in $d$, and then extract a complete passage $p_{final} = d[st : st + l_p]$ starting from $st$ with the length of $l_p$ tokens. For scenarios involving multiple prefixes, we also by default take the first one where the prefix appears. For final passage selection, we compute passage scores and two-stage scores using the autoregressive scores of short-prefix generation.

## 4 EXPERIMENTS

In this section, we conduct comprehensive experiments at the page and passage levels, and on downstream tasks, verifying the effectiveness of our method. Additionally, we carry out further analysis and experiments.

### 4.1 EXPERIMENTAL SETUP

**Datasets** We conduct extensive experiments on 6 knowledge-sensitive tasks from the KILT benchmark (Petroni et al., 2021). These include open-domain question answering tasks such as NQ (Kwiatkowski et al., 2019), TriviaQA (Joshi et al., 2017), HotpotQA (Yang et al., 2018), and ELI5 (Fan et al., 2019), the fact-checking task FEVER (Thorne et al., 2018), as well as the open-domain dialogue system WoW (Dinan et al., 2018). Since the KILT dataset lacks a publicly available test set, all experiments are conducted on its validation set. For specific details on the dataset, please refer to Appendix A.2. We evaluate the performance of page-level and passage-level retrieval tasks, as well as downstream tasks.

**Evaluation Metrics** We utilize R-Precision as the evaluation metric for our page-level retrieval tasks. For passage-level retrievals in datasets such as NQ, TriviaQA, and HotpotQA, we calculate the percentage of retrieved passages that contain at least one gold-standard answer, known as Answer in Context. In other datasets, we measure the percentage of retrieved passages that encompass at least one gold-standard entity, referred to as Entity in Context. For downstream tasks, we employ various metrics: Exact Match (EM) scoring for NQ, TriviaQA, and HotpotQA; Rouge-L for ELI5; accuracy for FEVER; and F1 score for WoW.

**Baselines** We evaluate several retrieval models. For unsupervised retrieval models, we compare traditional sparse retrieval models such as BM25 [2] (Robertson et al., 2009) and dense retrieval models such as Contriever (Izacard et al., 2022). We also compare the highly trained dense retrieval model DPR [3] (Karpukhin et al., 2020). These models all adopt the passage segmentation from the official KILT as the retrieval data source. We take the top 1 passage retrieved by the model as the reference context and input it into a large language model, which reads the related passage and then responds to the downstream tasks.

**Implementation Details** We select the 7b and 13b versions of the open-source large language models Llama (Touvron et al., 2023a) and Llama2 (Touvron et al., 2023b) for conducting experiments on generative retrieval and downstream tasks. We merge the passage segments from KILT into complete documents, serving as the data source for generative retrieval. The complete documents are of arbitrary length. In generative retrieval, we consistently employ a beam search generation strategy. During the first stage of generation, the beam size is set to 15, and we construct an FM-index with the top $k = 2$ documents. In the second phase, the beam size is set to 10, the length of short prefix generation is $l_{p_s} = 16$, and we extract passage with a token length of $l_p = 150$ as the final reference. The weight for the two-stage weighting method is set to $\alpha = 0.9$. Greedy decoding is employed for all downstream tasks. The prompts used in the experiments can be found in Appendix A.3. All experiments are conducted on Tesla A100 40G GPUs.

### 4.2 EXPERIMENTAL RESULTS

#### 4.2.1 RETRIEVAL RESULTS

The results for page-level retrieval are depicted in Table 1. Our method LLM2GR achieves the best R-precision scores of 57.77, 48.70, 83.69 and 57.63 on the NQ, HotpotQA, FEVER, and WoW datasets respectively when using Llama2 13b as the generative retrieval model. This greatly surpasses the performance of sparse retrieval BM25 and dense retrieval Contriever in the zero-shot scenario. It also presents strong competitiveness against the fully trained DPR method, particularly on the WoW and FEVER datasets, marking 27.08 and 31.01 points improvement respectively. Additionally, the general enhancement in performance is observed with the advancement from Llama to

---

[2]We implement BM25 retrieval using the repository https://github.com/castorini/pyserini

[3]We conduct experiments using the trained DPR model and preprocessed vector indexes from the https://github.com/facebookresearch/KILT repository.

| Method | Open-domain QA | | | | Fact Check. | Dial. |
| | NQ | TriviaQA | HotpotQA | ELI5 | FEVER | WoW |
|---|---|---|---|---|---|---|
| Contriever | 34.72 | 34.28 | 26.14 | 11.02 | 55.64 | 29.67 |
| BM25 | 26.33 | 31.78 | 41.30 | 6.83 | 52.09 | 28.78 |
| DPR⋆ | 54.74 | 45.68 | 25.46 | **16.19** | 56.61 | 26.62 |
| LLM2GR(Llama 7b) | 52.56 | 55.35 | 43.88 | 14.27 | 75.46 | 42.21 |
| LLM2GR(Llama 13b) | 51.53 | **56.62** | 46.09 | 13.80 | 74.05 | 28.32 |
| LLM2GR(Llama2 7b) | 56.26 | 56.52 | 46.20 | 14.60 | 77.27 | 49.64 |
| LLM2GR(Llama2 13b) | **57.77** | 54.41 | **48.70** | 15.00 | **83.69** | **57.63** |

Table 1: Page-level retrieval results, measured by R-Precision. ⋆ indicates that full data training has been conducted. Bold data in the table represents the best results, while underlined data indicates the second-best results.

| Method | Open-domain QA | | | | Fact Check. | Dial. |
| | NQ | TriviaQA | HotpotQA | ELI5 | FEVER | WoW |
| | Answer in Context | | | Entity in Context | | |
|---|---|---|---|---|---|---|
| Contriever | 19.28 | 37.21 | 11.16 | 12.48 | 40.48 | 45.15 |
| BM25 | 23.65 | 58.87 | 29.45 | 12.01 | 58.33 | 50.36 |
| DPR⋆ | **47.94** | 66.60 | 20.29 | 14.40 | 41.22 | 45.38 |
| LLM2GR(Llama 7b) | 34.72 | 55.96 | 24.43 | 14.93 | 54.67 | 53.70 |
| LLM2GR(Llama 13b) | 36.55 | 61.28 | 26.43 | 15.46 | 53.49 | 45.28 |
| LLM2GR(Llama2 7b) | 38.03 | 62.87 | 27.48 | **16.92** | 56.19 | 57.86 |
| LLM2GR(Llama2 13b) | 40.82 | **68.20** | **30.04** | 15.06 | **58.42** | **63.43** |

Table 2: Passage-level retrieval results, measured by Answer in Context and Entity in Context of top 1 evidence passage. ⋆ indicates full data training. Bold data in the table represents the best results, while underlined data indicates the second-best results.

Llama2 and the increase in model size, indicating the correlation between the efficacy of generative retrieval and the capabilities of the underlying large language models. We also observe that some phenomena of the inverse scaling law occur when using Llama for page retrieval on WoW, but these phenomena disappear in Llama2. A more powerful language model can mitigate the phenomena of the inverse scaling law to some extent. Owing to resource limitations, experiments on larger models are deferred to future work.

The results of passage-level retrieval are shown in Table 2. Our method LLM2GR also achieves the best scores of 68.20, 30.04, 58.42 and 63.43 on the TriviaQA, HotpotQA, FEVER and WoW datasets respectively when using Llama2 13b as the generative retrieval model. We note that in passage-level generative retrieval, the improvement compared to the DPR method has decreased relative to page-level retrieval. This indicates potential for optimization in activating large language models to generate more detailed and lengthier passages, presenting a greater challenge compared to generating shorter titles. Notably, DPR excels in the NQ dataset, which is related to its training data format. Interestingly, in the HotpotQA dataset, BM25 remains competitive, surpassing dense retrieval methods, possibly due to the longer questions in this dataset leading to more vocabulary overlap. LLM2GR shows significant advancement on the FEVER and WoW datasets, demonstrating the adaptability of large language models in generating passages for different tasks.

### 4.2.2 DOWNSTREAM TASK RESULTS

The results of the downstream tasks are presented in Table 3. Under the Llama2 13b setting, LLM2GR achieves the best scores of 72.94, 78.79, and 14.77 on TriviaQA, FEVER, and WoW tasks respectively, verifying its significant efficiency and broad potential for application. On the NQ dataset for open-domain question answering, although DPR performs exceptionally well after full-data training, LLM2GR also presents highly competitive performance. On the other hand, in

| Method | Open-domain QA | | | | Fact Check. | Dial. |
| | NQ | TriviaQA | HotpotQA | ELI5 | FEVER | WoW |
| | | EM | | R-L | ACC | F1 |
|---|---|---|---|---|---|---|
| Contriever(Llama 7b) | 13.64 | 55.76 | 16.70 | 20.48 | 61.35 | 14.00 |
| Contriever(Llama 13b) | 21.92 | 68.82 | 20.84 | 20.33 | 71.57 | 13.66 |
| Contriever(Llama2 7b) | 23.30 | 67.33 | 20.34 | 20.35 | 51.85 | 13.69 |
| Contriever(Llama2 13b) | 24.78 | 69.25 | 20.34 | 20.71 | 73.61 | 13.96 |
| BM25(Llama 7b) | 15.16 | 59.30 | 22.70 | 20.45 | 60.85 | 13.81 |
| BM25(Llama 13b) | 22.98 | 70.34 | 26.96 | 20.29 | 72.52 | 13.76 |
| BM25(Llama2 7b) | 25.03 | 69.10 | 26.59 | 20.14 | 51.91 | 13.79 |
| BM25(Llama2 13b) | 25.84 | 71.49 | **27.23** | 20.48 | 77.54 | 14.02 |
| DPR⋆(Llama 7b) | 22.91 | 60.37 | 19.66 | 20.68 | 59.49 | 13.98 |
| DPR⋆(Llama 13b) | 30.81 | 71.02 | 23.39 | 20.22 | 70.55 | 14.11 |
| DPR⋆(Llama2 7b) | 31.27 | 70.80 | 22.98 | 20.55 | 51.43 | 14.01 |
| DPR⋆(Llama2 13b) | **33.49** | 72.68 | 23.13 | **20.75** | 75.27 | 14.17 |
| LLM2GR(Llama 7b) | 18.65 | 55.95 | 20.43 | 20.73 | 60.27 | 14.36 |
| LLM2GR(Llama 13b) | 26.79 | 69.90 | 25.00 | 20.17 | 72.31 | 13.87 |
| LLM2GR(Llama2 7b) | 28.13 | 67.85 | 24.82 | 20.43 | 51.43 | 14.16 |
| LLM2GR(Llama2 13b) | 31.69 | **72.94** | 26.13 | 20.61 | **78.79** | **14.77** |

Table 3: Downstream task results. ⋆ indicates that the retrieval model has been trained with full data. Bold data in the table represents the best results, while underlined data indicates the second-best results.

| Method | NQ | TriviaQA | HotpotQA | NQ | TriviaQA | HotpotQA |
| | | R-Precision | | | Answer in Context | |
|---|---|---|---|---|---|---|
| LLM2GR | 57.77 | 54.41 | 48.70 | 40.82 | 68.20 | 30.04 |
| w/o weight | 51.22 | 49.23 | 48.70 | 39.06 | 66.86 | 28.88 |
| w/o SPGL | 55.30 | 51.50 | 48.70 | 37.43 | 64.64 | 26.18 |
| w/o first stage | 32.22 | 24.87 | 23.36 | 36.27 | 63.33 | 24.16 |

Table 4: The ablation study results on the NQ, TriviaQA, and HotpotQA datasets are presented, with the left half showing the R-Precision for page-level retrieval, and the right half showing the Answer in Context for passage-level retrieval. We compared the performance differences without weighted scores, without SPGL, and without the first stage title retrieval.

the TriviaQA and HotpotQA datasets, due to the length of the questions, BM25 achieves excellent performance by obtaining more vocabulary overlap, yet LLM2GR still achieves comparable or better performance in most cases. Contriever, without supervised training, performs relatively poorly across all tasks, emphasizing the crucial role of supervised training in enhancing the performance of dense retrieval models. Notably, as our LLM2GR method employs the same model in both the retrieval stage and the downstream task stage, the improvement in downstream tasks increases even more as the base model size and performance enhance, i.e., the sum of improvements in retrieval results and reading abilities.

## 4.3 ABLATION STUDY

In this subsection, we perform ablation studies to compare the methods without weighted scores (w/o weight), without Short Prefix Generation and Localization (w/o SPGL), and without first stage title retrieval (w/o first stage). The results are displayed in Table 4.

For the method without weighted scores, solely relying on the scores from second stage passage generation results in a decrease in both page and passage retrieval performance, underscoring the

significance of considering scores from both stages. The model, by considering title scores, can select evidence passages from the correct documents. However, solely depending on passage scores sometimes leads to the selection of incorrect reference passages. Additional results on weighted $\alpha$ selection are available in Appendix A.4.

Regarding method without SPGL, generating longer segments has a minor impact on page retrieval but more significantly affects passage retrieval performance. This outcome is somewhat counterintuitive, suggesting that shorter segments may already encompass key information that large language models deem as evidence, while longer passages introduce redundancy and noise, thereby reducing effectiveness. Notably, when utilizing Llama2 13b as the generation retrieval model, generating complete passages takes around 600 minutes on the NQ dataset, while short prefix generation only requires 150 minutes, significantly lowering the time cost. However, considering dense retrieval takes about 20 minutes, further optimization for generation retrieval speed is still crucial. More experiments on prefix length are in Appendix A.5.

For the method without first stage title retrieval, there's a further drop in passage-level retrieval, which significantly impacts page retrieval performance. This indicates considerable limitations and improvement opportunities in using large language models for direct passage generation. The capability of solely prompting large language models to generate fine-grained passages is limited, making the first stage title identifier generation retrieval vital.

### 4.4 FURTHER ANALYSIS

**Large Language Model after General Fine-Tuning** We also experiment with the Vicuna model (Chiang et al., 2023), post general fine-tuning, and the Llama2-chat model (Touvron et al., 2023b), refined through human feedback reinforcement learning. These general fine-tunings do not significantly improve large language models' performance in generative retrieval. This may be due to the discrepancy in paradigms between the fine-tuning data and generative retrieval, coupled with most knowledge being acquired during pre-training. Further enhancement in model performance could potentially be realized by creating more diverse generative retrieval instruction tuning data. Detailed results are available in Appendix A.6.

**The Impact of Few-Shot** We explore incorporating few-shot prompts in the passage generation stage and observe their impact on generative retrieval performance. This approach yields a minor improvement only on the HotpotQA dataset, while showing a slight decrease on NQ and TriviaQA. Importantly, adding more few-shot examples significantly slows down generation speed. This indicates that, while few-shot prompts present a potential improvement pathway, achieving more effective prompting methods still requires extensive exploration. Detailed results are available in Appendix A.7.

**Memory Usage Analysis** The dense retrieval methods such as Contriever and DPR both require over 60 GB of memory usage. In contrast, sparse retrieval methods use far less memory, requiring only 17 GB. The LLM2GR method utilizes FM-index and Trie indexes, where encoding and storing all documents in advance with FM-index only needs 8 GB, and storing Trie of all title identifiers requires merely 25 MB, which is almost negligible. Our method of storage is more memory-efficient compared to both sparse and dense retrieval methods.

## 5 CONCLUSION AND FUTURE WORK

This paper introduces a method named **LLM2GR**, which employs large language models for generative retrieval, flexibly applicable to various knowledge-sensitive tasks. Mimicking the human habit of searching for information, we initially prompt the large language model to identify relevant document pages, and then locate the corresponding reference passages from these pages. Additionally, through beam search constrained by Trie and FM-index, we ensure that the content generated by the large language model is a subset of existing text. This method can be flexibly paired with various open-source large language models, simplifying the retrieval steps and providing new guidance for the wider application of large language models. In the future, we consider enhancing the performance of large language models in generating relevant passages through instruction tuning, applying this method to more retrieval domains, exploring ways to inject new document knowledge into large language models, and integrating multi-hop reasoning into generative retrieval.

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

# A APPENDIX

## A.1 CONSTRAINED DECODING METHODS

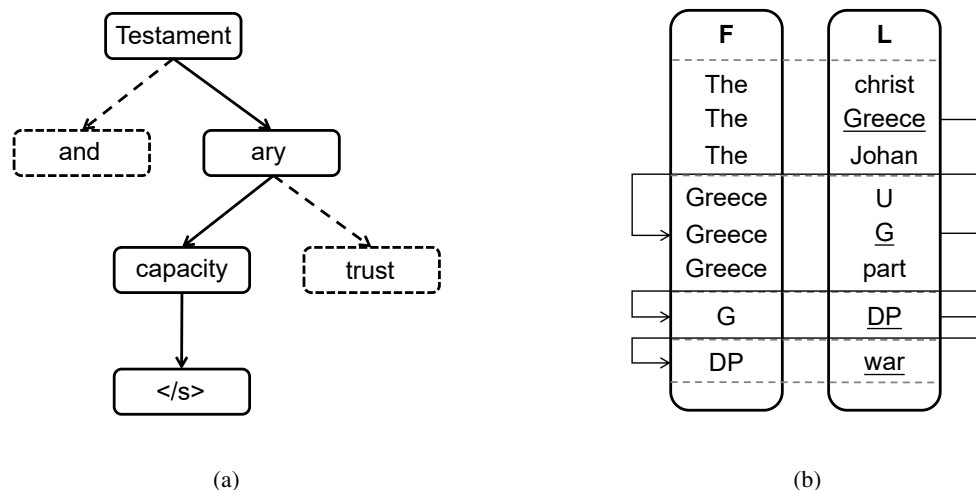

(a)                                                                 (b)

Figure 3: (a) Shows the process of a large language model generating title identifiers using a prefix tree. (b) Shows the process of a large language model generating passage prefixes in a document set via FM-index.

### A.1.1 TRIE

The Trie (Cormen et al., 2022), also known as a dictionary tree or prefix tree, is a tree-like data structure used to store an associative array where the keys are usually strings. Unlike a binary search tree, keys in a Trie are not stored directly within the nodes; instead, they are determined by the node's position in the tree. All descendants of a node have the same prefix, associated with the string corresponding to that node.

The overall process during constrained decoding using a Trie is shown in Figure 3a. Taking the generation of the title "Testamentary Capacity" as an example, the large language model first selects "Testament" from the set of token strings that start all titles. Subsequently, we can obtain the set of token strings {and, ary} following the string "Testament". After the large language model selects "ary", we get the prefix string "Testamentary", and finally continue to select new strings from the next set of token strings until the end-of-sequence token is encountered, ceasing generation.

### A.1.2 FM-INDEX

The FM-index (Ferragina & Manzini, 2000) is a data structure used for text retrieval that can store text efficiently with linear space complexity and support fast substring search operations. It is constructed based on the Burrows-Wheeler Transform (BWT) (Burrows, 1994). BWT is a method that converts a string into a form that is easy to compress. Given a string, BWT produces a transformed string through the following steps: generate all cyclic shifts of the string, sort all these shifts lexicographically, take the last character of each sorted shifted string to form a new string, which is the BWT result. For example, for the string "CABAC", the process of building the FM-index is as follows:

$$
\begin{array}{ccccccc}
\mathbf{F} & & & & & \mathbf{L} \\
\$^6 & C & A & B & A & C^5 \\
A^2 & B & A & C & \$ & C^1 \\
A^4 & C & \$ & C & A & B^3 \\
B^3 & A & C & \$ & C & A^2 \\
C^5 & \$ & C & A & B & A^4 \\
C^1 & A & B & A & C & \$^6
\end{array}
$$

where $ is a special string termination token, the numbers in the upper right corner of the letters in the F and L columns are the corresponding position index numbers. The FM-index explicitly stores two main parts: the F column and the L column. The F column is the lexicographically sorted characters of the transformed string, and the L column is the result of BWT. In addition, it stores additional position information to recover the original string from the BWT result. When we want to query a substring, the FM-index starts from the last character of the substring, using the information in the F column and the L column to gradually narrow down the possible position range until the exact position of the substring is determined or the substring is determined to be non-existent.

The overall process during constrained decoding using FM-index is shown in Figure 3b. Considering the generated prefix "The Greece GDP warrants are not technically bonds as investors do" for example, it first starts from the string "The" generated from all corpus, and gets its corresponding L column string set {christ, Greece, Johan}. After "Greece" is selected by the large language model, we can get the next set {U, G, part}, and continue the iteration until reaching the set maximum prefix length to stop generating.

## A.2 DATASET DETAILS

- Natural Questions (NQ) (Kwiatkowski et al., 2019) is constructed from real anonymized aggregated queries submitted to the Google search engine, with answers being snippets from manually annotated Wikipedia articles.

- TriviaQA (Joshi et al., 2017) comprises a set of questions and answers initially crawled from knowledge question-answering and quiz websites.

- HotpotQA (Yang et al., 2018) contains a set of question-answer pairs based on Wikipedia, requiring multiple-step reasoning over multiple Wikipedia pages to answer each question.

- ELI5 (Fan et al., 2019) is a large-scale corpus for long-form question answering, consisting of questions and answers from the Reddit forum "Explain Like I'm Five" (ELI5), which require detailed and in-depth responses to open-ended questions.

- FEVER (Thorne et al., 2018) is one of the largest datasets for fact-checking, used to determine whether a statement is supported or refuted based on textual sources.

- Wizard of Wikipedia (WoW) (Dinan et al., 2018) is a task that requires intelligent agents in open-domain dialogues to demonstrate knowledge usage. The dialogues are directly grounded in knowledge retrieved from Wikipedia.

All experiments are tested using the public validation set as divided in the official KILT. Additional details of the datasets are presented in Table 5.

| Dataset | Task | Input Format | Output Format | Size |
|---------|------|--------------|---------------|------|
| NQ | Open Domain QA | Question | Extractive | 2837 |
| HotpotQA | Open Domain QA | Question | Short Abstractive | 5600 |
| TriviaQA | Open Domain QA | Question | Extractive | 5359 |
| ELI5 | Open Domain QA | Question | Long Abstractive | 1507 |
| FEVER | Fact Checking | Claim | Classification | 10444 |
| WoW | Dialogue | Conversation | Long Abstractive | 3054 |

Table 5: Additional details of the datasets.

## A.3 PROMPTS

In this subsection, we introduce the prompts used in the first stage title identifier generation retrieval, the second stage passage generation retrieval, and the downstream tasks.

### A.3.1 PROMPTS FOR THE FIRST STAGE

- Open-domain Question Answering: "Question: {}\n\nThe Wikipedia article corresponding to the above question is:\n\nTitle:"

- Fact Verification: "Claim: {}\n\nThe Wikipedia article corresponding to the above claim is:\n\nTitle:"

- Open-domain Dialogue System: "Conversation: {}\n\nThe Wikipedia article corresponding to the above conversation is:\n\nTitle:"

### A.3.2 PROMPTS FOR THE SECOND STAGE

- Open-domain Question Answering: "Question: {}\n\nThe Wikipedia paragraph to answer the above question is:\n\nAnswer:"

- Fact Verification: "Claim: {}\n\nThe Wikipedia paragraph to support or refute the above claim is:\n\nAnswer:"

- Open-domain Dialogue System: "Conversation: {}\n\nThe Wikipedia paragraph to answer the above conversation is:\n\nAnswer:"

### A.3.3 PROMPTS FOR READING COMPREHENSION

- Open-domain Question Answering (NQ, TriviaQA, HotpotQA): "Refer to the passage below and answer the following question with just a few words.\nPassage: {}\nQ: {}\nA: The answer is"

- Open-domain Question Answering (ELI5): "Refer to the passage below and answer the following question in detail.\nPassage: {}\nQ: {}\nA:"

- Fact Verification: "background: {}\nclaim: {}\nQ: Is the claim true or false?\nA:"

- Open-domain Dialogue System: "background: {}\n{}\n"

### A.4 EXPERIMENTAL RESULTS FOR DIFFERENT VALUES OF ALPHA

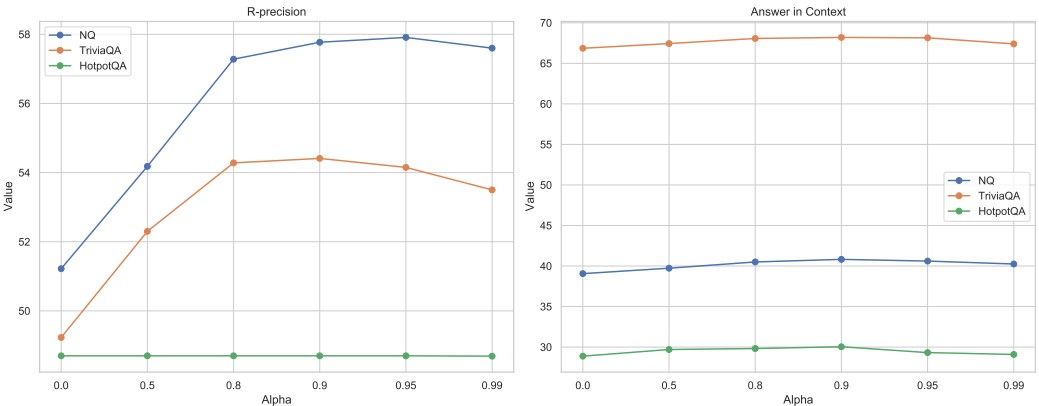

Figure 4: Experimental results for page-level and passage-level retrieval on the NQ, TriviaQA, and HotpotQA datasets with $\alpha$ set to {0.0,0.5,0.8,0.9,0.95,0.99} are presented.

In Figure 4, we conduct a comparison of experimental results with different values of $\alpha$. When $\alpha = 0.0$, it is equivalent to not having a two-stage weighting method, and relying solely on the scores generated by the second-stage paragraph results in the selection of suboptimal references. As $\alpha$ increases, the model sees improvements in both page-level and passage-level retrieval, demonstrating the importance of the first stage document scores for final reference selection. However, when $\alpha$ reaches 0.95 and continues to increase, the final performance actually decreases to a certain extent, indicating that a balance between the two needs to be struck to achieve better results.

### A.5 EXPERIMENTAL RESULTS FOR DIFFERENT PREFIX LENGTHS

In Figure 5, we conduct experiments generating different numbers of prefix tokens. We observe that longer prefix lengths do not bring about additional performance improvement; rather, they lead to

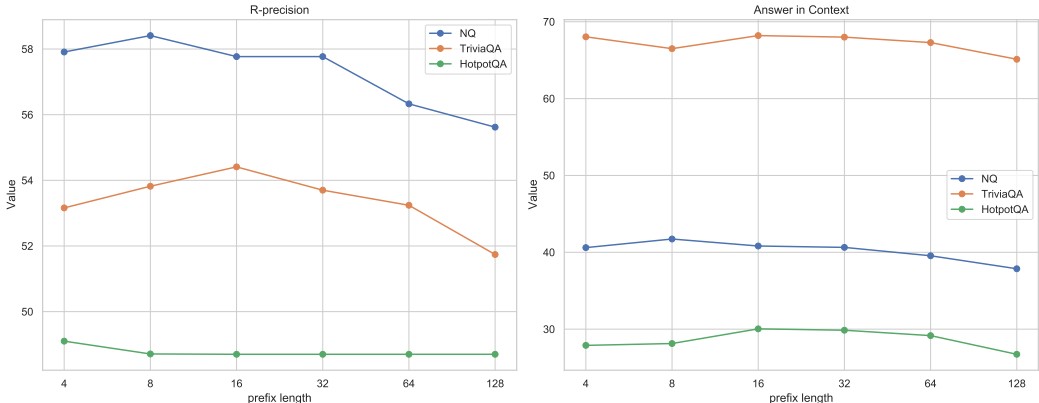

Figure 5: Experimental results for page-level and passage-level retrieval with varying numbers of prefix tokens $l_{p_s}$ set to {4,8,16,32,64,128} on the NQ, TriviaQA, and HotpotQA datasets are presented.

| Method | NQ | TriviaQA | HotpotQA | NQ | TriviaQA | HotpotQA |
|---|---|---|---|---|---|---|
| | | R-Precision | | | Answer in Context | |
| LLM2GR(Llama 7b) | 52.56 | 55.35 | 43.88 | 34.72 | 55.96 | 24.43 |
| LLM2GR(Vicuna 1.3 7b) | 48.47 | 47.99 | 40.79 | 35.28 | 56.41 | 23.75 |
| LLM2GR(Llama 13b) | 51.53 | 56.62 | 46.09 | 36.55 | 61.28 | 26.43 |
| LLM2GR(Vicuna 1.3 13b) | 52.73 | 46.61 | 43.41 | 36.55 | 67.29 | 26.63 |
| LLM2GR(Llama2 7b) | 56.26 | 56.52 | 46.20 | 38.03 | 62.87 | 27.48 |
| LLM2GR(Llama2-chat 7b) | 3.31 | 1.12 | 0.98 | 4.09 | 3.97 | 3.43 |
| LLM2GR(Vicuna 1.5 7b) | 50.76 | 51.73 | 41.23 | 34.16 | 55.98 | 24.43 |
| LLM2GR(Llama2 13b) | 57.77 | 54.41 | 48.70 | 40.82 | 68.20 | 30.04 |
| LLM2GR(Llama2-chat 13b) | 1.94 | 1.60 | 1.55 | 6.38 | 7.93 | 4.71 |
| LLM2GR(Vicuna 1.5 13b) | 52.24 | 56.34 | 45.90 | 37.22 | 63.24 | 27.14 |

Table 6: Experimental results of the model, after general fine-tuning on the NQ, TriviaQA, and HotpotQA datasets, are presented. The left side shows the R-Precision at the page-level, while the right side displays the Answer in Context at the passage-level.

a decline in retrieval ability. Existing large language models still perform better in generating and retrieving shorter segments; longer segments introduce additional noise, resulting in performance degradation. However, overly short prefixes also fail to contain sufficient information, leading to an inability to select the needed passages.

## A.6 EXPERIMENTAL RESULTS OF LARGE LANGUAGE MODELS AFTER GENERAL FINE-TUNING

In Table 6, we compare the performance of models in generative retrieval after supervised fine-tuning (Vicuna 1.3 and Vicuna 1.5) and human feedback reinforcement learning (Llama2-chat). It is observed that the Vicuna models, after supervised fine-tuning, do not exhibit further improvements in generative retrieval; in fact, the performance slightly declines. This suggests that the memorization of document knowledge is mostly accomplished during the pretraining phase, and further enhancements may require specific fine-tuning data for generative retrieval paradigms. In contrast, the performance of models subjected to human feedback reinforcement learning significantly decreases, unable to fully realize generative retrieval. We notice that models trained with human feedback reinforcement learning often start their outputs with polite phrases such as "Sure!", which affects the model distribution and consequently leads to the failure of generative retrieval.

## A.7 EXPERIMENTAL RESULTS UNDER FEW-SHOT PROMPTS

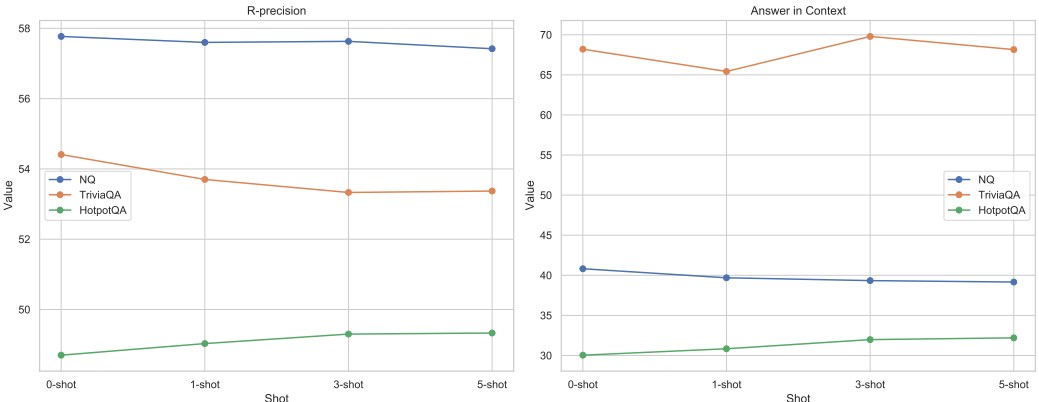

Figure 6: Experimental results for page-level and passage-level retrieval under few-shot prompts of {0,1,3,5}-shot on the NQ, TriviaQA, and HotpotQA datasets are presented.

In Figure 6, we compare the experimental results after incorporating few-shot prompts in the passage retrieval stage. The use of more sample prompts only results in partial improvements on the HotpotQA dataset, and the improvement levels off as the number of samples increases from 3 to 5. On the NQ and TriviaQA datasets, there are no further improvements and even a slight decline is observed. There is still a need to explore more effective ways of prompting.

## A.8 EXPERIMENTAL RESULTS OF SELECTING DIFFERENT NUMBERS OF FIRST STAGE DOCUMENTS

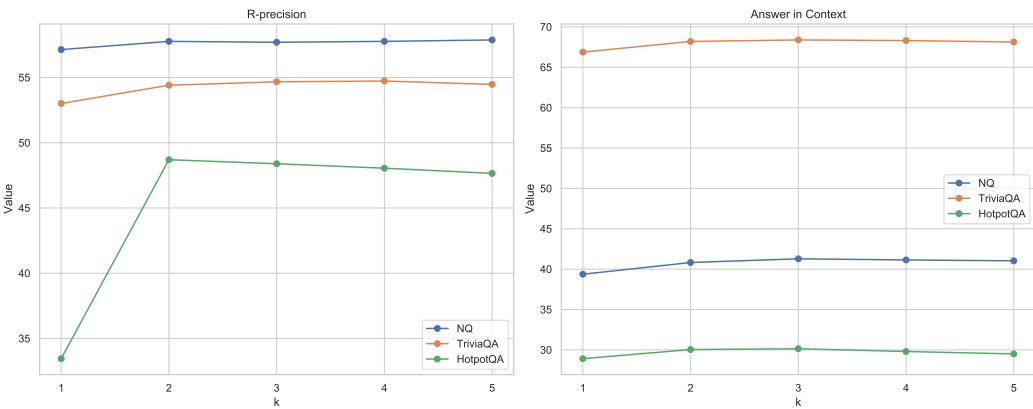

Figure 7: Experimental results for page-level and passage-level retrieval on the NQ, TriviaQA, and HotpotQA datasets with the number of documents selected in the first stage, $k$, set to {1,2,3,4,5} are presented.

In Figure 7, we conduct experiments comparing the effects of selecting different numbers of first stage documents, denoted as $k$. We observe that $k$ does not significantly impact the final performance, as the necessary effective passages are usually contained within the initial few documents. The subpar performance observed when $k = 1$ for HotpotQA can be attributed to the dataset requiring two documents to calculate R-Precision.

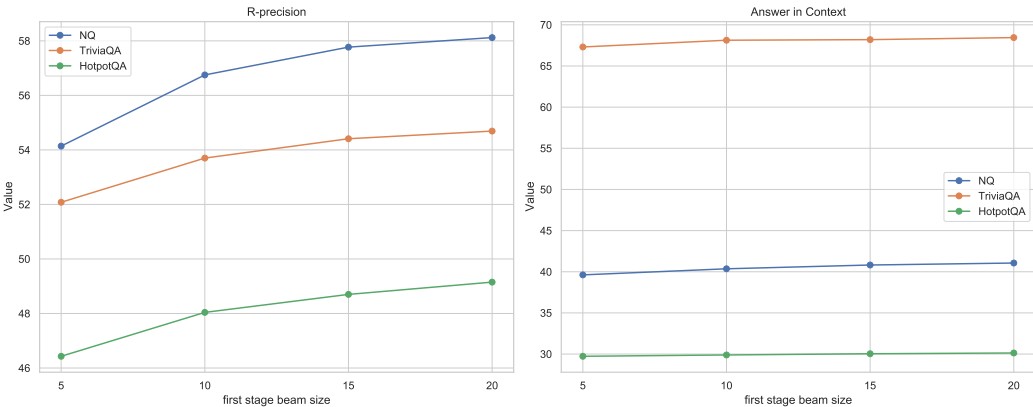

Figure 8: Experimental results for page-level and passage-level retrieval with different beam search sizes set to {4,8,16,32,64,128} in the first stage on the NQ, TriviaQA, and HotpotQA datasets are presented.

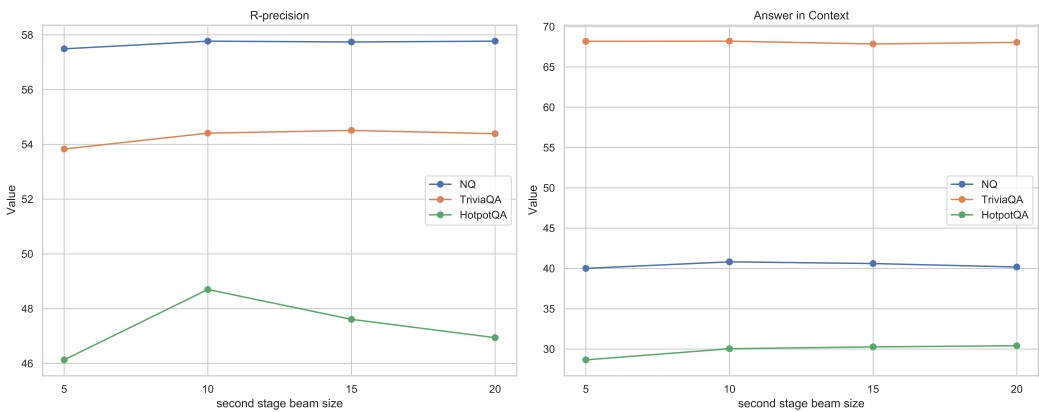

Figure 9: Experimental results for page-level and passage-level retrieval with different beam search sizes set to {4,8,16,32,64,128} in the second stage on the NQ, TriviaQA, and HotpotQA datasets are presented.

## A.9    EXPERIMENTAL RESULTS WITH DIFFERENT BEAM SEARCH SIZES

Figures 8 and 9 respectively show the impact of different beam search sizes on the results in the first and second stages. For the first stage title identifier generative retrieval, a larger beam size can yield better page-level retrieval results, thus slightly improving the effectiveness of the second stage passage retrieval. However, in the second stage passage generative retrieval, the improvement brought by a larger beam size is not significant and can even result in a slight decline, which might be due to the additional noise introduced by a larger beam size.

## A.10    CASE STUDY

In Tables 7 through 12, we showcase retrieval cases on the NQ, TriviaQA, and HotpotQA datasets using the Gold Standard, BM25, and LLM2GR methods. LLM2GR has obtained results containing the answers in Tables 7, 9, and 11 by generating passages more aligned with the questions. Table 8 presents a biology question; although the passage retrieved by LLM2GR did not contain the annotated answer, it provided a more detailed description of the location and process where chymotrypsin cleaves peptide bonds. However, Tables 10 and 12 illustrate instances where LLM2GR experienced retrieval failures. This occurs as merely generating a related prefix at times cannot ensure that the subsequent part will definitely contain the answer, resulting in a passage that is only generally re-

lated. Ensuring the flexibility of generative retrieval while taking into account more subsequent information still necessitates further exploration. Notably, the questions in the TriviaQA and Hot-potQA datasets are longer compared to the NQ dataset, which allows for more vocabulary overlap between the questions and passages, enhancing the effectiveness of sparse retrieval for BM25.

| |
|---|
| **Query:** what is the legal age for marriage in australia |
| **Gold Standard:** Part II (s.10-21) deals with the marriageable age and the marriage of minors. In the original 1961 Act, marriageable age was set at 16 for females and  18  for males. However, under section 12 of the original 1961 Act a female 14 or 15 years or a male 16 or 17 years could apply to the court for permission to marry. The marriageable age was equalised in 1991 by the "Sex Discrimination Amendment Act 1991", which raised the marriageable age of females to  18 . |
| **BM25:** marriage. Before 2004, there was no definition of marriage in the 1961 Act, and instead the common law definition used in the English case "Hyde v Hyde" (1866) was considered supreme. Though s.46(1) of the Act required celebrants to explain the legal nature of marriage in Australia to a couple as "the union of a man and a woman to the exclusion of all others, voluntarily entered into for life", these words were descriptive or explanatory, rather than outlining what constituted a legally valid marriage in |
| **LLM2GR:** Marriage in Australia is regulated by the federal "Marriage Act 1961" (Cth) , which applies uniformly throughout Australia (including its external territories) to the exclusion of all state laws on the subject. Australian law recognises only monogamous marriages, being marriages of two people, including same-sex marriages, and does not recognise any other forms of union, including traditional Aboriginal marriages, polygamous marriages or concubinage. The marriage age for marriage in Australia is  18  years, but in "unusual and exceptional circumstances" a person aged 16 or 17 can marry with parental consent |

Table 7: In the NQ dataset, the Gold Standard, BM25, and LLM2GR reference passage for the query "what is the legal age for marriage in Australia" are provided. The parts containing the answer are highlighted with a  grey  background.

| |
|---|
| **Query:** where does cleavage of the peptide bond by chymotrypsin occur |
| **Gold Standard:** "In vivo", chymotrypsin is a proteolytic enzyme (serine protease) acting in the  digestive systems  of many organisms. It facilitates the cleavage of peptide bonds by a hydrolysis reaction, which despite being thermodynamically favorable, occurs extremely slowly in the absence of a catalyst. The main substrates of chymotrypsin are peptide bonds in which the amino acid N-terminal to the bond is a tryptophan, tyrosine, phenylalanine, or leucine. Like many proteases, chymotrypsin also hydrolyses amide bonds "in vitro", a virtue that enabled the use of substrate analogs such as N-acetyl-L-phenylalanine p-nitrophenyl amide for enzyme assays. |
| **BM25:** 149, producing $\alpha$-chymotrypsin (which is more active and stable than $\pi$-chymotrypsin). The resulting molecule is a three-polypeptide molecule interconnected via disulfide bonds. "In vivo", chymotrypsin is a proteolytic enzyme (serine protease) acting in the  digestive systems  of many organisms. It facilitates the cleavage of peptide bonds by a hydrolysis reaction, which despite being thermodynamically favorable, occurs extremely slowly in the absence of a catalyst. The main substrates of chymotrypsin are peptide bonds in which the amino acid N-terminal |
| **LLM2GR:** Chymotrypsin cleaves peptide bonds by attacking the unreactive carbonyl group with a powerful nucleophile, the serine 195 residue located in the active site of the enzyme, which briefly becomes covalently bonded to the substrate, forming an enzyme-substrate intermediate. Along with histidine 57 and aspartic acid 102, this serine residue constitutes the catalytic triad of the active site. These findings rely on inhibition assays and the study of the kinetics of cleavage of the aforementioned substrate, exploiting the fact that the en |

Table 8: In the NQ dataset, the Gold Standard, BM25, and LLM2GR reference passage for the query "where does cleavage of the peptide bond by chymotrypsin occur" are provided. The parts containing the answer are highlighted with a  grey  background.

| |
|---|
| **Query:** Which expression is associated with the sinking of the HMS Birkenhead at Gansbaai near Cape Town, South Africa, in Febuary 1852? |
| **Gold Standard:** Only 193 of the estimated 643 people on board survived, and the soldiers' chivalry gave rise to the unofficial "women and children first" protocol when abandoning ship, while the "Birkenhead drill" of Rudyard Kipling's poem came to describe courage in face of hopeless circumstances. |
| **BM25:** HMS "Birkenhead, also referred to as HM Troopship "Birkenhead or Steam Frigate "Birkenhead", was one of the first iron-hulled ships built for the Royal Navy. She was designed as a steam frigate, but was converted to a troopship before being commissioned. She was wrecked on 26 February 1852, while transporting troops to Algoa Bay at Danger Point near Gansbaai, 87 miles (140 kilometres) from Cape Town in the Cape Colony. There were not enough serviceable lifeboats for all the passengers, and the soldiers |
| **LLM2GR:** The sinking of the "Birkenhead" is one of the earliest maritime disaster evacuations during which the concept of "women and children first" is known to have been applied."Women and children first" subsequently became standard procedure in relation to the evacuation of sinking ships, in fiction and in life. The term ""Birkenhead" drill" became defined as courageous behaviour in hopeless circumstances and appeared in Rudyard Kipling's 1893 tribute to the Royal Marines, "Soldier an' Sailor Too": To take your chance in the thick of a rush, with firing all about, Is nothing so bad when you've |

Table 9: In the TriviaQA dataset, the Gold Standard, BM25, and LLM2GR reference passage for the query "Which expression is associated with the sinking of the HMS Birkenhead at Gansbaai near Cape Town, South Africa, in Febuary 1852?" are provided. The parts containing the answer are highlighted with a grey background.

| |
|---|
| **Query:** Who was the first winner of 'I'm A Celebrity, Get Me Out Of Here'? |
| **Gold Standard:** The first series of I'm a Celebrity...Get Me Out of Here! was broadcast on ITV from 25 August to 8 September 2002. Ant & Dec presented the main show on ITV, whilst Louise Loughman hosted the spin-off show "I'm a Celebrity...Get Me Out of Here! NOW!" on ITV2. The winner of this series was radio DJ Tony Blackburn. |
| **BM25:** The first series of I'm a Celebrity...Get Me Out of Here! was broadcast on ITV from 25 August to 8 September 2002. Ant & Dec presented the main show on ITV, whilst Louise Loughman hosted the spin-off show "I'm a Celebrity...Get Me Out of Here! NOW!" on ITV2. The winner of this series was radio DJ Tony Blackburn. The show began with 8 celebrity contestants. The contestants take part in daily trials to earn food All ratings are taken from the UK |
| **LLM2GR:** I'm a Celebrity...Get Me Out of Here! is a British reality TV series in which a number of celebrities live together in a jungle environment for a number of weeks, competing to be crowned "King" or "Queen of the Jungle". The show was originally created in the United Kingdom by the factual programmes department of ITV's then London franchise, London Weekend Television and developed by a team including James Allen, Natalka Znak, Brent Baker and Stewart Morris. The first episode aired on 25 August 2002 . It is now produced by ITV Studios and has been licensed globally to countries including the United States |

Table 10: In the TriviaQA dataset, the Gold Standard, BM25, and LLM2GR reference passage for the query "Who was the first winner of 'I'm A Celebrity, Get Me Out Of Here'?" are provided. The parts containing the answer are highlighted with a grey background.

| |
|---|
| **Query:** 2014 S/S is the debut album of a South Korean boy group that was formed by who? |
| **Gold Standard:** 2014 S/S is the debut album of South Korean group WINNER. It was released on August 12, 2014 by the group's record label, YG Entertainment . The members were credited for writing the lyrics and composing the majority of the album's songs. |
| **BM25:** S is a South Korean project group consisting of three members: Kangta, Lee Ji-hoon and Shin Hye-sung. The group debuted in 2003, under the SM Entertainment label. After 11 years, they released and promoted another mini-album in 2014. In 2003, S released their first album "Fr.In.Cl", which stands for "Friends in Classic". In 2014, after more than a decade in hiatus, the group released their second mini-album "Autumn |
| **LLM2GR:** 2014 S/S is the debut album of South Korean group WINNER. It was released on August 12, 2014 by the group's record label, YG Entertainment . The members were credited for writing the lyrics and composing the majority of the album's songs. The members produced the majority of the music for this album themselves, with the help of other producers such as Choice 37, B.I., Airplay, and others. The album was highlighted for incorporating elements generally absent from K-pop releases, including hints of acoustic |

Table 11: In the HotpotQA dataset, the Gold Standard, BM25, and LLM2GR reference passage for the query "2014 S/S is the debut album of a South Korean boy group that was formed by who?" are provided. The parts containing the answer are highlighted with a grey background.

| |
|---|
| **Query:** Who is the fictional head of a British Secret Service division and for which a one-time missionary was the inspiration for? |
| **Gold Standard:** Charles Fraser-Smith (26 January 1904 – 9 November 1992) was an author and one-time missionary who is widely credited as being the inspiration for Ian Fleming's James Bond quartermaster Q . During World War II, Fraser-Smith worked for the Ministry of Supply, fabricating equipment nicknamed " Q -devices" (after Q -ships) for SOE agents operating in occupied Europe. Prior to the war, Fraser-Smith had worked as a missionary in North Africa. After the war he purchased a dairy farm in Bratton Fleming, Devon, where he died in 1992. |
| **BM25:** Q is a fictional character in the James Bond films and film novelisations. Q (standing for Quartermaster), like M, is a job title rather than a name. He is the head of Q Branch (or later Q Division), the fictional research and development division of the British Secret Service. The use of letters as pseudonyms for senior officers in the British Secret Intelligence Service was started by its first director Captain Sir Mansfield George Smith-Cumming (1859-1923) who signed himself with a C written in |
| **LLM2GR:** Ian Fleming created the fictional character of James Bond as the central figure for his works. Bond is an intelligence officer in the Secret Intelligence Service, commonly known as MI6. Bond is known by his code number, 007, and was a Royal Naval Reserve Commander. Fleming based his fictional creation on a number of individuals he came across during his time in the Naval Intelligence Division and 30 Assault Unit during the Second World War, admitting that Bond "was a compound of all the secret agents and commando types I met during the war". Among those types were his brother, Peter, who had been involved in behind-the-lines operations in Norway and Greece during the war. |

Table 12: In the HotpotQA dataset, the Gold Standard, BM25, and LLM2GR reference passage for the query "Who is the fictional head of a British Secret Service division and for which a one-time missionary was the inspiration for?" are provided. The parts containing the answer are highlighted with a grey background.

