# OpenReview forum: "Generative Retrieval with Large Language Models"
_ICLR.cc/2024/Conference — Submitted to ICLR 2024_

### Official Review · Reviewer_KiwB · 2023-10-31

**Soundness:** 3 good
**Presentation:** 3 good
**Contribution:** 3 good
**Rating:** 5
**Confidence:** 4

**Summary:**

The paper presents a constrained decoding strategy to limit the generative space of large language models, effectively transforming them into retrievers without requiring additional training. This retrieval process is two-fold: initially, the large model generates titles corresponding to relevant documents based on input questions. Subsequently, the model generates paragraphs within these documents that contain the sought-after answers. To alleviate the computational burden associated with generating full paragraphs, the authors introduce a technique where the language model produces prefixes to quickly locate relevant paragraphs. This approach substantially reduces the number of tokens generated by the LLM, thereby achieving computational efficiency. The final output of LLM is generated  based on the comprehension of the identified content obtained above. The authors validate their method through experiments on six KILT benchmarks tailored for knowledge-sensitive tasks. The proposed method demonstrates superior performance across the majority of these benchmarks.

**Strengths:**

- This study offers a novel approach to transforming large language models into search retrievers by ingeniously leveraging constrained decoding. This technique guides the models to generate document titles or content within predefined boundaries, achieving retrieval capabilities without necessitating extra training, specialized retrieval models, or text chunking processes.
- Addressing the second-stage bottleneck where the Language Model's (LLM) generation of paragraph text is time-consuming, the paper introduces the Short Prefix Generation and Location (SPGL) method. This significantly improves the retrieval speed, reducing it from 600 minutes to 150 minutes.
- The paper is exceptionally well-articulated and clear in its presentation.

**Weaknesses:**

- Despite the authors' efforts to enhance computational efficiency through the use of shorter prefixes and the Knuth-Morris-Pratt (KMP) algorithm for passage location, the proposed LLM2GR model still necessitates at least three forward computations.  This computational burden remains a significant hindrance to its practical applicability.
- The comparison between LLM2GR and existing methods such as Dense Retriever (DPR), BM25, and Contriever raises questions of fairness. Specifically, LLM2GR employs a considerably more powerful LLM for generative tasks, which inherently skews the comparison with baselines of lesser model capacity. Using an LLM for retrieval is not inherently novel. A more equitable evaluation might involve integrating LLMs with existing retrieval methods—such as DPR—to compare generative capabilities. For instance, one could employ DPR for retrieval and then use an LLM for the generative task based on the retrieved results, thus providing a fairer baseline for comparison. If the proposed two-staged method can still deliver better results, thus demonstrating the effectiveness of the proposed method.
- Writing Errors: There are typographical errors in the manuscript, such as in Section 2.3 where the phrase "generate related contexts for "retrieval,"" is incorrectly punctuated. Additionally, the usage of quotation marks throughout the paper is inconsistent.
- A significant limitation of the proposed method is its dependence on the universal knowledge gained during the pretraining stage. This raises questions about the method's applicability in scenarios that go beyond the scope of Wikipedia or in specialized vertical fields such as medicine or economics. The paper does not address how the method could be adapted or if it remains effective when applied to these more specialized domains. This limitation is a critical factor that constrains my enthusiasm for strongly recommending the paper for acceptance.
- While the paper convincingly demonstrates that Large Language Models (LLMs) can be used for retrieval, it lacks an in-depth discussion on why LLMs can perform this task effectively. Is it because the models have been exposed to the documents during the pre-training phase?

**Questions:**

- In section 3.3，given a short text prefix $p_s$ of $l_{p_s}$ tokens, the KMP algorithm is used to determine the start position $st$ of $p_s$, then a complete passage is extracted by catching starting from $st$ with the length of $l_p$ tokens. Here a question arises that if the text prefix $p_s$ is equal to the final retrieved passage $p_final = d[st : st + l_p]$?
- Regarding the slow generation speed of the LLM, the authors should explore techniques like 8-bit or 4-bit quantization to assess whether such methods can accelerate the model's performance to an acceptable level, while also evaluating the trade-off in retrieval accuracy.

---

> ### Author Response · Authors · 2023-11-19
> **Response to Reviewer KiwB**
>
> Thank you very much for your careful review!
>
> **W1: The trade-off of two-stage method and speed consideration**
>
> The two-stage method does require two forward passes, but by generating short snippets instead of one long passage at a time, we can still greatly reduce the time consumption compared to single-stage methods. Due to the current difficulty for LLMs to directly recall fine-grained passages, we adopted the trade-off of using a two-stage approach.
>
> **W2: Differences from combining existing LLMs and traditional retrieval like DPR**
>
> There are many studies trying to combine LLM generation with traditional methods like DPR, which is similar to using retrieval tools. Our method aims to explore a completely new paradigm of performing retrieval solely based on the capabilities of LLMs themselves, without needing external tools. We think these two lines of work are actually parallel, and prompting LLMs to mimic human recall itself to accomplish retrieval has unique significance. We also reproduced the query2doc (Wang et al.,2023) method of generating for retrieval enhancement using the same hyperparameters on NQ and WOW datasets.
>
> | Method | Page-level NQ | Page-level WOW | passage-level NQ | passage-level WOW |
> |-|-|-|-|-|
> | bm25 | 26.33 | 28.78 | 23.65 | 50.36 |
> | query2doc+bm25 | 47.55 | 48.49 | 41.42 | 62.02 |
> | LLM2GR | 57.77 | 57.63 | 40.82 | 63.43 |
>
> The query2doc method enhances retrieval by prompting LLM to generate virtual document. We see significant improvements over bm25. However, our method can still achieve better page-level results using LLMs alone, and competitive passage-level results.
>
> In addition, query2doc takes ~430 minutes on NQ while our LLM2GR only needs 150 minutes.
>
> **W3: Writing issues**
>
> Thank you for pointing out the writing issues, we will make corresponding modifications!
>
> **W4: Other domain**
>
> Currently our method is only experimented on Wikipedia data, because LLMs are mostly trained on general domain data like Wikipedia during pre-training, while their performance is inferior on some vertical domains which they are not sufficiently trained on. We fully agree on the importance of applying LLM retrieval to various vertical domains, but this is a limitation of our current method. Continued exploration is needed on incorporating new domain documents more effectively, and instruction tuning to activate stronger recall.
>
> **W5: Memorization during pre-training**
>
> We think the main reason our method works is because during pre-training LLMs are exposed to large amounts of documents, so like humans after reading many books, LLMs also remember some document details. This allows us to activate recall of relevant snippets directly through prompting, and ensure factuality via indexed constraints.
>
> **Q1: Prefix length**
>
> Here we set the prefix length much smaller than the final length, 16 vs 150. So few cases will have identical prefix and final lengths. When the prefix is at the end, we will still extract from the end back to the prefix, which does not impact performance significantly.
>
> **Q2: Quantization methods**
>
> We fully agree that model quantization can greatly improve the speed of LLM generative retrieval, but since our paper focuses on the retrieval approach itself, we did not specifically consider quantized models. Analyzing the combination in the future is an interesting research direction! But model quantization is orthogonal to our short prefix generation method, and can potentially further improve our current approach.

---

### Official Review · Reviewer_M43s · 2023-10-31

**Soundness:** 3 good
**Presentation:** 2 fair
**Contribution:** 2 fair
**Rating:** 5
**Confidence:** 4

**Summary:**

This paper introduces a new generative retrieval method called LLM2GR, which prompts LLama (7b,13b) to generate title identifiers, then further filter out the paragraphs from the retrieved documents by prompting the LM again to generate short prefixes using constrained decoding and find documents that correspond prefix. They evaluate proposed methods on the KILT benchmark and compare the retrieval performance with BM25, Contriever, and DPR baselines.

I have several major concerns about this paper, which are detailed in Weaknesses. In summary, I think
- The technical novelty of this paper is still unclear
- They should have evaluated their proposed methods' effectiveness with other generative retrieval methods e.g., generating URLs or generative paragraphs in terms of performance and efficiency.
- The main contributions or advantages (efficiency, not limited by URLs) of the methods are not fully supported by the experimental results.

**Strengths:**

- This paper introduces a new generative retrieval method that does not rely on additional reranking components
- Instead of genereating full paragraphs, the proposed method only generate short prefix and search documents, which can improve inference time efficiency.

**Weaknesses:**

The major three concerns I have are as follows:
1. Limited technical contributions and novelties
2. Missing important baselines
3. Unsupported claims

**1. Limited technical contributions and novelties**

Generative retrieval methods have been actively studied, and generating full evidence paragraphs (Yu et al., 2023), URLs (Ziems et al., 2023), Wikipedia titles (De Cao et al., 2020), or substrings (Bevilacqua et al., 2022) have been explored. Bevilacqua et al. (2022) generate substrings, which are similar to short prefix generations, and use the FM-Index to constrain the autoregressive generations. It is not discussed in depth how this work is different from Bevilacqua et al. (2022). One difference might be prompting an LM  rather than fine-tuning. Yet, if the novelty is mostly prompting a larger LM rather than fine-tuning a smaller LM, I think it is not a sufficient contribution for ICLR.

**2. Missing important baselines**

Despite that the proposed method is a new generative retrieval method, none of the baselines is generative retrieval. Several prior works including Bevilacqua et al. (2022) or Yu et al. (2023) are tested on KILT or open-domain QA datasets and are open-source, and I wonder why the authors did not directly compare their method with such prior work.


**3. Unsupported claims**

Several advantages mentioned in the introduction or the related work section have not been validated by experimental results. For instance,

- *The process of generating complete passages can be overly time-consuming, posing a significant drawback to their practical implementation. To address this issue, we propose a novel approach termed Short Prefix Generation and Location (SPGL).*

If the authors claim this method is more efficient for practical application by generating less (but searching over FM Index & locating prefixes via KMP), the authors should benchmark inference time latency. Given the complex pipeline of the proposed method and active research of fast inference (e.g., speculative decoding, paged attentions), I am unsure whether this method is indeed practical by skipping longer generations or taking more time due to the expensive search process. Related to the previous point discussed above, the authors should benchmark other generative retrieval methods and quantitatively show the proposed method's effectiveness.

- *Recently, Ziems et al. (2023) have proposed using the large language model GPT-3 to retrieve documents by generating document URLs. However, this method can only be applied to documents retrievable by URLs*

The proposed method is also limited to the pre-processed Wikipedia corpus as it matches the generated sequences with all titles in DB, which means that the proposed method's search space is also limited by existing Wikipedia titles. For me the difference between the form of URLs v.s. titles does not matter if you only test on Wikipedia, as in Wikipedia converting title into URLs can be achieved by simple text postprocessing most of the time.

Yu, Wenhao, et al. "Generate rather than retrieve: Large language models are strong context generators." ICLR (2023).

**Questions:**

- Did you evaluate other generative retrieval model performance as well as inference latency?
- Di you compare the inference time latency of non-genreative retrieval methods and your method?

---

> ### Author Response · Authors · 2023-11-19
> **Response to Reviewer M43s**
>
> Thank you very much for your careful review!
>
> **W1: Technical novelty**
>
> Our method explores a completely new retrieval paradigm - generating determined snippets directly by utilizing the capability of large language models to mimic human recall, without any segmentation of long texts or extra models and training. Constrained decoding solves the hallucination problem of directly generating using LLMs in Yu et al. (2023). It also addresses the inferior performance of Ziems et al. (2023) on smaller LLMs.
>
> The key difference from SEAL(Bevilacqua et al., 2022) is that we rely on the capability of LLMs themselves to determine the start positions of fine-grained snippets, instead of generating part of pre-segmented passages to finish retrieval, which introduces much more difficulty. But we believe this retrieval idea will have greater flexibility in the future.
>
> **W2: Comparison with other generative retrieval baseline**
>
> Since all components of Yu et al. (2023) are implemented using closed APIs, direct comparison with our method using open-source LLMs like llama would be very unfair. For llm-url we reproduced results using the same hyperparameters and llama2 13b on NQ and WOW datasets.
>
> | Method | Page-level NQ | Page-level WOW | passage-level NQ | passage-level WOW |
> |-|-|-|-|-|
> | bm25 | 26.33 | 28.78 | 23.65 | 50.36 |
> | DPR | 54.74 | 26.62 | 47.94 | 45.38 |
> | SEAL | 64.15 | 50.49 | 44.94 | 62.08 |
> | llm-url+bm25 | - | - | 19.67 | 43.09 |
> | LLM2GR | 57.77 | 57.63 | 40.82 | 63.43 |
>
> The trained SEAL method has been greatly improved compared to dense retrieval methods, showing the effectiveness of generative retrieval through internal model interactions. Note that both SEAL and DPR require extensive training, while our method can achieve competitive results in zero-shot settings by prompting LLMs.
>
> The llm-url method generates Wikipedia URLs using LLMs. Since it is hard to directly map to KILT benchmark documents, we only provided passage-level results. We found llama2 13b can only generate valid urls with efficacy of 39.5% and 45.2% on the two datasets. This makes it hard to generate sufficient accurate URLs to achieve effective results when using smaller LLMs. Our method can achieve competitive results using smaller open-source LLMs with prompting and constrained decoding, which will facilitate the real-world application of generative retrieval.
>
> **W3: Using faster decoding methods**
>
> We think using faster decoding methods is actually orthogonal to our current approach. Combined with faster LLM inference in the future, our existing method will lead to faster generative retrieval. Since our method focuses more on the retrieval performance itself, we did not incorporate fast model inference. However, designing specialized accelerating algorithms for generative retrieval is an interesting future direction.
>
> **Q1/Q2: Inference latency**
>
> As analyzed in the ablation study in Section 4.3, our method takes about 150 minutes on the NQ dataset, dense retrieval takes about 20 minutes, and the generative retrieval method SEAL needs about 11 minutes, llm-url takes about 1700 minutes to generate URLs. Currently due to the expensive cost of LLMs, the time is still far more than traditional fine-tuning of small models. Further optimizing the speed of generative retrieval is crucial.

---

> > ### Comment · Reviewer_M43s · 2023-11-22
> > **Thank you for your response.**
> >
> > Thank you for your response. Given that I’m happy increase my score to 5. I’m still uncertain that the paper provides enough contributions and notable differences from prior work. I understand the decoding speeds are orthogonal. My point in the original point is that while generating snippets is motivated by the latency of generating the whole passage or longer sub sequences, due to the complexity of pipeline, the proposed method may even slower than methods that generating the full sequences.

---

> > > ### Author Response · Authors · 2023-11-23
> > > **Thank you for your reply!**
> > >
> > > Thank you very much for your reply! We believe that exploring the combination of large language models' inherent capabilities with indexing methods for retrieval is a very interesting direction. The current multi-stage approach is also a kind of trade-off. When people cannot accurately remember content, they might build hierarchical structures to aid memory. Allowing large language models to choose the granularity of these hierarchies themselves is also a potential direction. However, for now, since the length of the content generated in both stages is relatively short, it offers a significant speed advantage over directly generating a longer paragraph.
> > >
> > > Thank you again for your detailed reply, it's very helpful to us!

---

### Official Review · Reviewer_i5R3 · 2023-11-01

**Soundness:** 3 good
**Presentation:** 3 good
**Contribution:** 2 fair
**Rating:** 5
**Confidence:** 4

**Summary:**

This paper proposes LLM2GR, a two-stage generative retrieval based on LLM under the zero-shot setting, which first generates title identifiers to obtain top k documents, and then generates directly the content of passages whose length is around 150 to 200 tokens, among a set of passages that belong to the top k documents retrieved at the first stage. Two scores resulting from both stages are linearly interpolated to finally rank passages. Furthermore, this paper proposes a short prefix generation and location (SPGL) to efficiently locate and extract a relatively long passage. In SPGL, instead of generating a full content of a passage, its prefix is first generated and then the remaining content of the passage is just located based on KMP string matching algorithm. Under SPGL, the score of the second stage is now replaced with the score of generating only prefix (not the score of the full content). Experiment results using LLMs of llama 7b~13b show that the proposed two-stage generative retrieval often outperforms the fully-finetuned DPR method in TriviaQA, HotpotQA, FEVER, and WoW datasets, demonstrating its promisingness. Ablation studies show that the first stage is vital and the prefix-based score is better than the score using the full content.

**Strengths:**

- The paper proposes the initial work of LLM for generative retrieval under zero-shot setting. While generative retrieval has been popularly studied, the work presents the new exploration of LLM focusing on the zero-shot setting, showing that the zero-shot generative retrieval under LLM is promising, although the proposed method is technically simple.
- Experiment results are largely interesting, confirming that the zero-shot generative retrieval using LLM shows improved performances over the fully-finetuned DPR, while the reranking method is not compared. Ablation study includes the comparison between the major variants.
- The short prefix generation and location (SPGL) is further proposed, as an effective way to address the computational overhead of large model to generate long documents. While SPGL is also simple, it is reasonably designed and makes the further improvement.

**Weaknesses:**

- It seems that the proposed two-stage method is technically not very attractive. Under LLM, a simpler design like generating title identifier and prefix in a single stage would be preferred. The kind of simpler method needs to be considered and adopted as the baseline for the comparison.
- It seems that the baselines are weak. In the fully-finetuned case, not only DPR but also the reranking method needs to be provided, as the LLM generates the passages based on a stronger attention-based interaction between a question and a document, than the DRP that uses the inner product between dense vectors.
- The retrieval performance was only compared with previous dense and sparse retrieval methods, lack of comparison with recent generative retrieval methods.
- In the proposed method, only zero-shot settings are presented. Few-shot settings of LLM need to also be provided.
Under few-shot prompts, other LLM-based method such as query2doc could be compared.
https://arxiv.org/pdf/2303.07678.pdf
- Other popular datasets are not considered for evaluation, such as MS MARCO, TREC DL 19, and TREC DL 20. To convincingly validate the proposed zero-shot retrieval of LLM, the work needs to provide a full comparison on more IR collections.
- The construction of FM-index after the first stage seems to be done per a query. Its computational overhead is not fully discussed. Why FM-index is not constructed over the full documents, towards a manner of restricting a full FM-index to the only restricted top-k documents.
-  The proposed SPGL approach first generates short prefix of the document and then matches the prefix with document fragments using KMP algorithm. It is much faster than directly generating the whole document using LLM, however, whether this approach has a speed advantage over other generative retrieval works was not discussed. Many generative retrieval works also do not directly generate the documents, such as in SEAL [1], where n-grams of documents are generated first and then using a scoring function to determine the documents. The reviewers would like to know if SPGL is faster compared to other generative retrieval works.
- In section 3.3, the selection method of the unique documents (from document set) and prefix (from multiple prefixes) is naïve. Although in the vast majority of cases the “top k document” was unique in this dataset, the method of “default choice” does not have any ranking or selection capabilities in the case of extending this work to larger corpus or practical applications.

**Questions:**

- The proposed two-stage method needs to be compared with a single-stage method that generates a title identifier and prefix of a passage. Why a single-stage method is not considered?
- What is the computational time of constructing the FM-index? Is it not possible to construct the FM-index over the full passages, and to use it in a restricted way on only top-k documents?
- In Table 1-2, only DPR is compared. What is the best performance of the existing works on the same collections? The reranking-based method needs to also be compared.
- In section 3.2, it was claimed that the “information loss” issue of text chunking for lengthy documents was overcame by the proposed method, however, this work actually generate a short prefix of the document, which is also less informative than the whole document or document chunk. The reviewer would like to see the authors' opinion on this issue.
- What is the memory cost of using prefix tree and FM-index structures? Does this work require significant use of additional memory to store these indexes?
-  The length of the short prefix l_(p_s ) was not explicitly given, the reviewer would like to know how the authors decide the length of short prefixes and how does the length of the short prefix contribute to the retrieval performance?

---

> ### Author Response · Authors · 2023-11-19
> **Response to Reviewer i5R3 (1/2)**
>
> Thank you very much for your careful review!
>
> **W1: Single-stage method**
>
> In the ablation experiment in Section 4.3 of our paper, we compared the single-stage generation method of large language model based generative retrieval, i.e. the method without the first stage. In this setting, we directly prompt the large language model to generate the required fine-grained passages.
>
> | Method | Page-level NQ | Page-level TriviaQA | Page-level HotpotQA | Passage-level NQ | Passage-level TriviaQA | Passage-level HotpotQA |
> |-|-|-|-|-|-|-|
> | LLM2GR | 57.77 | 54.41 | 48.70 | 40.82 | 68.20 | 30.04 |
> | W/o first stage | 32.22 | 24.87 | 23.36 | 36.27 | 63.33 | 24.16 |
>
> We found that in this case the performance of current large language models would drop dramatically. Current large language models have difficulty recalling key segments from a vast space, so we adopted this trade-off of using the two-stage method.
>
> **W2: Re-ranking methods**
>
> Re-ranking methods need to recalculate cross attentions between the retrieved top-k passages and the question, i.e. access each retrieved passage during inference. Our method, like dense retrieval, does not directly access passages during retrieval, but relies on parameters stored in the model, which is different from re-ranking and is a parallel method to dense retrieval. Therefore, we did not compare with further re-ranking methods, similar to previous generative retrieval methods like SEAL(Bevilacqua et al., 2022).
>
> **W3/W4: Comparison with other generative retrieval baselines and query2doc**
>
> We conducted experiments with SEAL(Bevilacqua et al., 2022) and reproduced LLM-URL(Ziems et al., 2023) and Query2doc(Wang et al.,2023) using the same hyperparameter settings with Llama2 13b on NQ and WOW datasets.
>
> | Method | Page-level NQ | Page-level WOW | Passage-level NQ | Passage-level WOW |
> |-|-|-|-|-|
> | bm25 | 26.33 | 28.78 | 23.65 | 50.36 |
> | DPR | 54.74 | 26.62 | 47.94 | 45.38 |
> | SEAL | 64.15 | 50.49 | 44.94 | 62.08 |
> | llm-url+bm25 | - | - | 19.67 | 43.09 |
> | query2doc+bm25 | 47.55 | 48.49 | 41.42 | 62.02 |
> | LLM2GR | 57.77 | 57.63 | 40.82 | 63.43 |
>
> We found that the trained SEAL method has been greatly improved compared to dense retrieval methods, demonstrating the effectiveness of generative retrieval through internal model interactions. Both SEAL and DPR methods have gone through extensive training, while our method can achieve competitive results in zero-shot settings by prompting large language models.
>
> The llm-url method generates Wikipedia URLs using large language models, but it is difficult to directly map to KILT benchmark documents, so we only provided passage-level experimental results. We found that when using the llama2 13b model, it is difficult to generate enough valid Wikipedia URLs to achieve effective results.
>
> The query2doc method enhances retrieval by prompting LLM to generate virtual document. We noticed a significant improvement over the bm25 method. However, our method can still achieve better results in page-level retrieval by solely utilizing the capabilities of LLMs, and achieve competitive retrieval results at the passage-level.
>
> In addition, the llm-url method took about 1700 minutes on the NQ dataset, while query2doc took about 430 minutes, both exceeding the 150 minutes consumed by our llm2gr method.
>
> **W5: Other datasets**
>
> Since our method aims to generate fine-grained snippet information without any pre-segmentation of passages using large language models, while MS MARCO, TREC DL 19 and TREC DL 20 consist of fixed pre-segmented passages without providing original documents and final answers like the KILT dataset, it is difficult to properly evaluate our proposed new retrieval scheme. Therefore, we did not choose these standard IR datasets. The variety of task types and large amount of test data in KILT make it a valid evaluation in our opinion.
>
> **W6: FM-index pre-building**
>
> In fact, our method uses pre-built FM-index in implementation, which takes about 8h to build the entire Wikipedia corpus. During generation retrieval, we can directly extract the corresponding index sections, so the time consumption is negligible.

---

> ### Author Response · Authors · 2023-11-19
> **Response to Reviewer i5R3 (2/2)**
>
> **W7: The main difference compared to generating short snippets in SEAL**
>
> The biggest difference between our method and the SEAL method is that we do not need any pre-segmentation of passages. The start positions of passages are also discovered autonomously by the LLMs. We aim to explore a more flexible new retrieval scheme, which is also more challenging.
>
> **W8: Selecting the first prefix**
>
> Since we select from limited top-k documents, and the short prefixes also have a certain length, this method can produce a unique result in most cases. However, we also noticed that when documents get longer and more top-k documents are obtained, more ambiguity may occur leading to some bias. We think this bias can be solved by combining with re-ranking methods in future work. Since our paper focuses on the retrieval part, we leave this to future work.
>
> **Q1: Single-stage method**
>
> Same as W1.
>
> **Q2: FM-index pre-building**
>
> Same as W6.
>
> **Q3: Other baselines**
>
> Same as W2-W4.
>
> **Q4: The information amount of generated short prefixes**
>
> Regarding the concern that generated short prefixes may lack information amount, we think this information can be obtained from the information stored in the parameters of the large language model. When the language model can accurately recall relevant snippets, a short prefix is enough to locate the target snippet. However, when the recall ability of the language model is not strong enough, we think generating multiple snippet parts or using the multi-stage method like our method can help improve recall.
>
> **Q5: Memory usage of storage**
>
> As analyzed in Section 4.4 on storage, dense retrieval methods require about 60G memory, sparse retrieval BM25 needs about 17G, while our method only needs 25mb for the title prefix tree, and 8G for the pre-built full Wikipedia FM-index. Our method has more efficient memory utilization compared to previous methods.
>
> **Q6: The impact of short prefix length**
>
> In Appendix A.5 we experimented the impact of different prefix lengths on performance. We observed that over-short prefixes may not contain enough information, and shorter lengths make it hard to locate a unique passage, leading to performance drop. When generating longer prefixes, there is also a performance drop, slightly counter-intuitive. We analyzed this may be because current large language models are still more suitable for generating and retrieving shorter snippets, and longer prefixs introduce additional noise, causing performance degradation.

---

### Official Review · Reviewer_7CYr · 2023-11-01

**Soundness:** 2 fair
**Presentation:** 2 fair
**Contribution:** 2 fair
**Rating:** 6
**Confidence:** 4

**Summary:**

This paper is about generative retrieval. The key claim of the authors is to simulate the process of humans finding relevant information. To this end, the authors have developed a framework that tries to mimic the search process adopted by humans. Here the key model is the language model. From Figure 1, the model developed comprises of two stages:
1. In stage 1, the model finds relevant documents given a query
2. The model then finds a reference passage in the relevant document.

The authors then expand upon their idea in Figure 2 where they depict the complete model.

The authors then conduct a series of experiments to demonstrate the effectiveness of their approach.

Just a minor comment to the authors: the title of the paper is very general. This paper is about passage retrieval. It would have been nice had the authors reflected it in their paper title. Generative information retrieval has now become a broad field and continues to grow.

**Strengths:**

One of the key strengths is that the paper focuses on both the effectiveness and efficiency of the task. In terms of effectiveness, the model, from the experimental results demonstrates that it improves upon the existing method. The paper also discusses efficiency issues with large language modelling approaches, especially in retrieval settings where it is important to retrieve the relevant documents as efficiently as possible.

Passage retrieval has been long studied in the information retrieval literature. The goal is to retrieve relevant passages rather than documents so that a user can find their relevant information as quickly as possible. While there are traditional approaches such as the BM25 used in the past, this paper develops a generative learning method that has become popular recently using the transformer architecture.

In the two-stage method, the model mainly uses the Wikipedia document collection to help retrieve relevant information for the user. The quantitative results demonstrate that the model improves upon existing methods.

**Weaknesses:**

The issue with this work is that in the two-stage approach, the model will propagate errors from one stage to the next. While the unified approach would be highly complicated, the authors must realise that this is the shortcoming of this work.

It must be also noted by the authors that they use only Wikipedia articles that might help improve the efficiency of the model. The medium and large-sized models usually use a large amount of document collections which surely helps encode plenty of information which might not be present in Wikipedia, this does not mean that other models are relatively less efficient than this model. What would have been nice had the authors discussed the pros and cons of using large datasets that go beyond Wikipedia?

**Questions:**

It would be nice if the authors addressed the concerns that I have raised in the weakness section of my comments.

---

> ### Author Response · Authors · 2023-11-19
> **Response to Reviewer 7CYr**
>
> Thank you very much for your careful review! I also appreciate your comments on the title!
>
> **Q1: Error propagation of the two-stage method**
>
> Firstly, regarding the concern about the two-stage method, this is indeed a kind of trade-off when the current large language model (LLM) finds it difficult to directly activate its generation of fine-grained passages. In the ablation experiment in section 4.3 and Table 4 of our paper, we found that:
>
> | Method | Page-level NQ | Page-level TriviaQA | Page-level HotpotQA | Passage-level NQ | Passage-level TriviaQA | Passage-level HotpotQA |
> | ------ | ------------- | ------------------- | ------------------- | ---------------- | ---------------------- | ---------------------- |
> | LLM2GR | 57.77 | 54.41 | 48.70 | 40.82 | 68.20 | 30.04 |
> | w/o first stage | 32.22 | 24.87 | 23.36 | 36.27 | 63.33 | 24.16 |
>
> When the first stage of generating title identifiers to locate the required documents is not present, the performance drops dramatically. The current large language models have difficulty recalling key segments from a vast space in a zero-shot setting, so we have adopted this trade-off.
>
> **Q2: Use of Wikipedia Corpus**
>
> As for the use of the Wikipedia corpus, this is because the current large language models use data from the general domain most widely, and Wikipedia is the most typical one. This allows the LLM to effectively remember the content from this part of the pre-training phase. However, for more specialized corpora, the LLM has not undergone sufficient training, causing the LLM to not be able to recall this part of the content well, leading to poor performance in vertical domains. We fully agree with the importance of applying LLM to generate fine-grained passages in various vertical domains, but the current method cannot achieve results as superior as those with Wikipedia data, which is a limitation of the current method. Continuous exploration is still needed for more effective incorporation of new domain knowledge documents and research on instruction tuning to activate recall capabilities.

---

### Author Response · Authors · 2023-11-19
**General response**

Thank you very much for the careful reviews, which are very helpful for improving our paper!

We believe humans have the capability to recall arbitrary fine-grained snippets and gradually reconstruct a complete original passage. Therefore, our paper explores a completely new method of locating fine-grained snippets by solely utilizing the capabilities of large language models themselves, and combines this with indexing methods to address the hallucination problem. Our method explores a new dimension different from current approaches using various tools, to further explore the intrinsic retrieval capabilities of large language models. At the same time, we are fully aware of the challenges of directly utilizing the language models themselves without any external tools to recall and generate fine-grained passages. But we also hope our method can provide new insights for leveraging the capabilities of large language models themselves and exploring more flexible and accurate retrieval systems in the future.

---

### Meta-Review · Area_Chair_JWTx · 2023-12-06

**Metareview:**

The paper presents a novel approach to transform large language models into retrievers through constrained decoding. The authors propose a two-stage method, LLM2GR, and introduce a technique to improve computational efficiency, the Short Prefix Generation and Location (SPGL). The method demonstrates superior performance across the majority of the KILT benchmarks.

However, the reviewers have raised several concerns. The technical novelty of the paper is unclear, and the authors have not compared their method with other generative retrieval methods. The main advantages of the method are not fully supported by the experimental results. The paper also lacks an in-depth discussion on why LLMs can perform retrieval tasks effectively.

The computational burden (at least three forward computations) of the proposed LLM2GR model remains a significant hindrance to its practical applicability. The comparison between LLM2GR and existing methods raises questions of fairness, as LLM2GR employs a considerably more powerful LLM for generative tasks.

**Justification For Why Not Higher Score:**

1. the technical novelty of the paper is unclear, and the authors have not compared their method with other generative retrieval methods. This makes it difficult to assess the true contribution and effectiveness of the proposed method.

2. the main advantages of the method, such as improved efficiency, are not fully supported by the experimental results.

3. the paper lacks an in-depth discussion on why large language models can perform retrieval tasks effectively. This is a crucial aspect that needs to be addressed to fully understand the implications of the proposed method.

**Justification For Why Not Lower Score:**

N/A

---

### Decision · Program_Chairs · 2024-01-16

Reject